# Open Bandit Dataset and Pipeline: Towards Realistic and Reproducible Off-Policy Evaluation

**Yuta Saito**[*]
Cornell University
ys552@cornell.edu

**Shunsuke Aihara**
ZOZO Research
shunsuke.aihara@zozo.com

**Megumi Matsutani**
ZOZO Research
megumi.matsutani@zozo.com

**Yusuke Narita**
Yale University
yusuke.narita@yale.edu

## Abstract

*Off-policy evaluation* (OPE) aims to estimate the performance of hypothetical policies using data generated by a different policy. Because of its huge potential impact in practice, there has been growing research interest in this field. There is, however, no real-world public dataset that enables the evaluation of OPE, making its experimental studies unrealistic and irreproducible. With the goal of enabling realistic and reproducible OPE research, we present *Open Bandit Dataset*, a public logged bandit dataset collected on a large-scale fashion e-commerce platform, ZOZOTOWN. Our dataset is unique in that it contains a set of *multiple* logged bandit datasets collected by running different policies on the same platform. This enables experimental comparisons of different OPE estimators for the first time. We also develop Python software called *Open Bandit Pipeline* to streamline and standardize the implementation of batch bandit algorithms and OPE. Our open data and software will contribute to fair and transparent OPE research and help the community identify fruitful research directions. We provide extensive benchmark experiments of existing OPE estimators using our dataset and software. The results open up essential challenges and new avenues for future OPE research.

## 1 Introduction

Interactive bandit systems such as personalized medicine and recommendation platforms produce log data valuable for evaluating and redesigning the system. For example, the logs of a news recommendation system records which news article was presented and whether the user read it, giving the system designer a chance to make its recommendations more relevant. Exploiting log bandit data is, however, more difficult than conventional supervised machine learning, as the result is only observed for the action chosen by the system, but not for all the other actions that the system could have taken. The logs are also biased in that they overrepresent the actions favored by the system. A potential solution to this problem is an A/B test, which compares the performance of counterfactual systems in an online environment. However, A/B testing counterfactual systems is often difficult because deploying a new policy is time- and money-consuming and entails the risk of failure. This leads us to the problem of *off-policy evaluation* (OPE), which aims to estimate the performance of a counterfactual (or evaluation) policy using only log data collected by a past (or behavior) policy. OPE allows us to evaluate the performance of candidate policies without implementing A/B tests and contributes to safe policy improvement. Its applications range from

---

[*]This work was done when YS was at Hanjuku-kaso Co Ltd, Tokyo, Japan.

35th Conference on Neural Information Processing Systems (NeurIPS 2021) Track on Datasets and Benchmarks.

contextual bandits [4, 25, 26, 32, 37, 38, 40, 43, 44, 45, 51] and reinforcement learning in the web industry [10, 17, 20, 28, 47, 48, 52] to other social domains such as healthcare [31] and education [29].

**Issues with current experimental procedures.** Although the research community has produced theoretical breakthroughs over the past decade, the experimental evaluation of OPE remains primitive. Specifically, it lacks a public benchmark dataset for comparing the performance of different estimators. Researchers often validate their estimators using synthetic simulation environments [20, 28, 50, 52]. A version of the synthetic approach is to modify multiclass classification datasets and treat supervised machine learning methods as bandit policies to evaluate the estimation accuracy of OPE estimators [8, 10, 18, 49, 51]. An obvious problem with these studies is that they are *unrealistic* because there is no guarantee that their simulation environment is similar to real-world settings. To solve this issue, some previous studies use proprietary real-world datasets [12, 14, 32, 33]. Because these datasets are not public, however, the results are *irreproducible*, and it remains challenging to compare existing estimators with new ideas in a fair manner. The lack of a public real-world benchmark makes it hard to identify critical research challenges and the bottleneck of the literature. This contrasts with other domains of machine learning, where large-scale open datasets, such as the ImageNet dataset [7], have been pivotal in driving objective progress [9, 13, 15, 16].

**Contributions.** Our goal is to implement and evaluate OPE in *realistic and reproducible* ways. To this end, we release *Open Bandit Dataset*, a set of logged bandit datasets collected on the ZOZOTOWN platform.[2] ZOZOTOWN is the largest fashion e-commerce platform in Japan. When the platform produced the data, it used Bernoulli Thompson Sampling (Bernoulli TS) and uniform random (Random) policies to recommend fashion items to users. The dataset thus includes a set of *two* logged bandit datasets collected during an A/B test of these bandit policies. Having multiple (at least two) log datasets is essential because it enables the evaluation of the estimation accuracy of OPE estimators as we describe in detail in Section 5.

In addition to the dataset, we implement *Open Bandit Pipeline*, an open-source Python software including a series of modules for implementing dataset preprocessing, policy learning methods, and OPE estimators. Our software provides a complete, standardized experimental procedure for OPE research, ensuring that performance comparisons are fair, transparent, and reproducible. It also enables fast and accurate OPE implementation through a single unified interface, simplifying the practical use of OPE.

Using our dataset and software, we perform comprehensive benchmark experiments on existing estimators. We implement this OPE experiment by using the log data of one of the policies (e.g., Bernoulli TS) to estimate the policy value of the other policy (e.g., Random) with each OPE estimator. We then assess the accuracy of the estimator by comparing its estimation with the policy value obtained from the data in an *on-policy* manner. Through the experiments, we showcase the utility of Open Bandit Dataset and Pipeline by using them to analyze the challenges that we face when we try applying OPE to real-world scenarios.

Our key contributions are summarized as follows:

- **Public Dataset**: We build and release *Open Bandit Dataset*, a set of *two* logged bandit data to enable realistic and reproducible research on OPE.
- **Software Implementation**: We implement *Open Bandit Pipeline*, an open-source Python software that helps practitioners utilize OPE to evaluate their bandit systems. It also helps researchers compare different OPE estimators in a standardized manner.
- **Benchmark Experiment**: We perform comprehensive benchmark experiments on existing OPE estimators and indicate critical challenges in future research.

## 2 Off-Policy Evaluation

### 2.1 Setup

We consider a general contextual bandit setting. Let $r \in [0, r_{\max}]$ denote a reward variable (e.g., whether a fashion item as an action results in a click). We let $x \in \mathcal{X}$ be a context vector (e.g., the

---

[2]https://corp.zozo.com/en/service/

user's demographic profile) that the decision maker observes when picking an action. We also let $a \in \mathcal{A}$ be a discrete action such as a fashion item in a recommender system. Rewards and contexts are sampled from unknown distributions $p(r \mid x, a)$ and $p(x)$, respectively. We call a function $\pi : \mathcal{X} \to \Delta(\mathcal{A})$ a *policy*. It maps each context $x \in \mathcal{X}$ into a distribution over actions, where $\pi(a \mid x)$ is the probability of taking action $a$ given context $x$. We describe some examples of such decision making policies in Appendix A.

Let $\mathcal{D} := \{(x_i, a_i, r_i)\}_{i=1}^n$ be logged bandit dataset with $n$ observations. $a_i$ is a discrete variable indicating which action in $\mathcal{A}$ is chosen for $i$. $r_i$ and $x_i$ denote the reward and the context observed for each data, respectively. We assume that the logged dataset is generated by *behavior policy* $\pi_b$ as

$$\{(x_i, a_i, r_i)\}_{i=1}^n \; \sim \; \prod_{i=1}^n p(x_i)\pi_b(a_i \mid x_i)p(r_i \mid x_i, a_i),$$

where each triplet is sampled independently from the product distribution. We sometimes use $\mathbb{E}_n[f] := n^{-1} \sum_{(x_i, a_i, r_i) \in \mathcal{D}} f(x_i, a_i, r_i)$ to denote the empirical expectation over $n$ observations in $\mathcal{D}$. We also use $q(x, a) := \mathbb{E}_{r \sim p(r|x,a)}[r \mid x, a]$ and $g(x, \pi) := \mathbb{E}_{a \sim \pi(a|x)}[g(x, a) \mid x]$ to define estimators.

## 2.2 Estimation Target and Estimators

We are interested in using logged bandit data to estimate the following *policy value* of any given *evaluation policy* $\pi_e$, which might be different from $\pi_b$:

$$V(\pi_e) := \mathbb{E}_{(x,a,r) \sim p(x)\pi_e(a|x)p(r|x,a)}[r].$$

An OPE estimator $\hat{V}$ estimates $V(\pi_e)$ using only $\mathcal{D}$ as $V(\pi_e) \approx \hat{V}(\pi_e; \mathcal{D})$. We define three standard estimators in the following.[3]

**Direct Method (DM).** DM [2] first estimates $q$ using a supervised machine learning model, such as random forest or ridge regression. It then plugs it in to estimate the policy value as

$$\hat{V}_{\mathrm{DM}}(\pi_e; \mathcal{D}, \hat{q}) := \mathbb{E}_n[\hat{q}(x_i, \pi_e)],$$

where $\hat{q}(x, a)$ is a reward estimator. If $\hat{q}(x, a)$ is accurate, DM also estimates the policy value accurately. If $\hat{q}(x, a)$ is inaccurate, however, the final estimator is no longer consistent. The model misspecification issue is problematic because the extent of misspecification cannot be easily quantified from data [10, 50].

**Inverse Probability Weighting (IPW).** To alleviate the issue with DM, IPW is often used [35, 40]. This estimator weighs the observed rewards by the importance weights as

$$\hat{V}_{\mathrm{IPW}}(\pi_e; \mathcal{D}) := \mathbb{E}_n[w(x_i, a_i)r_i],$$

where $w(x, a) := \pi_e(a \mid x)/\pi_b(a \mid x)$. When the behavior policy is known, IPW is unbiased and consistent. However, it can have a large variance, especially when the evaluation policy deviates significantly from the behavior policy.

**Doubly Robust (DR).** DR [8] combines DM and IPW as follows.

$$\hat{V}_{\mathrm{DR}}(\pi_e; \mathcal{D}, \hat{q}) := \mathbb{E}_n[\hat{q}(x_i, \pi_e) + w(x_i, a_i)(r_i - \hat{q}(x_i, a_i))].$$

DR mimics IPW to use a weighted version of rewards, but it also uses $\hat{q}$ as a control variate to decrease the variance. It preserves the consistency of IPW if either the importance weight or the reward estimator is consistent (a property called *double robustness*). Moreover, DR is *semiparametric efficient* when the reward estimator is correctly specified [32]. However, when it is misspecified, this estimator can have a larger asymptotic mean-squared-error than that of IPW [20].

## 3 Open-Source Dataset and Software

Motivated by the paucity of real-world datasets and implementations enabling the evaluation of OPE, we release the following open-source dataset and software.

---

[3]We define some other advanced estimators in Appendix B.

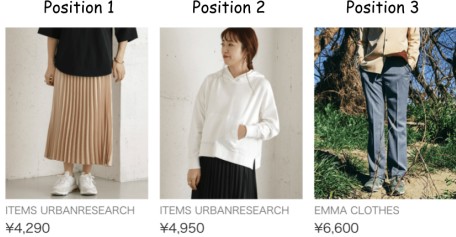

Figure 1: Fashion items as actions displayed in ZOZOTOWN recommendation interface.

Table 1: Statistics of Open Bandit Dataset

| Campaigns | Data Collection Policies | #Data | #Items | #Dim | CTR ($V(\pi)$) $\pm$95% CI | Relative-CTR |
|---|---|---|---|---|---|---|
| **ALL** | **Random** | 1,374,327 | 80 | 84 | 0.35% $\pm$0.010 | 1.00 |
| | **Bernoulli TS** | 12,168,084 | | | 0.50% $\pm$0.004 | 1.43 |
| **Men's** | **Random** | 452,949 | 34 | 38 | 0.51% $\pm$0.021 | 1.48 |
| | **Bernoulli TS** | 4,077,727 | | | 0.67% $\pm$0.008 | 1.94 |
| **Women's** | **Random** | 864,585 | 46 | 50 | 0.48% $\pm$0.014 | 1.39 |
| | **Bernoulli TS** | 7,765,497 | | | 0.64% $\pm$0.056 | 1.84 |

*Note*: Bernoulli TS stands for Bernoulli Thompson Sampling. **#Data** is the total number of user impressions observed during the 7-day experiment. **#Items** is the total number of items having a non-zero probability of being recommended by each policy. **#Dim** is the number of dimensions of the raw context vectors. **CTR** is the percentage of a click being observed in the log data, and this is the performance of the data collection policies in each campaign. 95% confidence interval (CI) of CTR is calculated based on the normal approximation of the Bernoulli sampling. **Relative-CTR** is CTR relative to that of the Random policy for the "ALL" campaign.

**Open Bandit Dataset.** Our open-source dataset is a set of *two* logged bandit datasets provided by ZOZO, Inc., the largest fashion e-commerce company in Japan. The company uses multi-armed bandit algorithms to recommend fashion items to users in their fashion e-commerce platform called ZOZOTOWN. We present examples of displayed fashion items in Figure 1. We collected the data in a 7-day experiment in late November 2019 on three "campaigns," corresponding to "ALL", "Men's", and "Women's" items, respectively. Each campaign randomly uses either the Random policy or the Bernoulli TS policy for each user impression. These policies select three of the candidate fashion items for each user. Figure 1 shows that there are three *positions* in our dataset. We assume that the reward (click indicator) depends only on the item and its position, which is a general assumption on the click generative process used in the web industry [27]. Under this assumption, we can apply the OPE setup in Section 2 to our dataset. We provide some statistics of the dataset in Table 1. The dataset is large and contains many millions of recommendation instances. Each row of the data has feature vectors such as age, gender, and past click history of the users. These feature vectors are hashed, thus the dataset does not contain any personally identifiable information. Moreover, the dataset includes some item-related features such as price, fashion brand, and item categories. It also includes the probability that item $a$ is displayed at each position by the data collection policies. This probability is used to calculate the importance weights.[4] We share the full version of our dataset at `https://research.zozo.com/data.html`.[5] Small-sized example data are also available at `https://github.com/st-tech/zr-obp/tree/master/obd`.

To our knowledge, our open-source dataset is the first to include logged bandit datasets collected by running *multiple* (at least two) different policies and the exact policy implementations used in real production, enabling ***realistic and reproducible evaluation of OPE*** for the first time. Indeed, Open Bandit Dataset is already actively used by multiple research papers to benchmark new bandit algorithms and OPE estimators [6, 22, 23, 39, 46].

---

[4] We computed the action choice probabilities by Monte Carlo simulations based on the policy parameters (e.g., parameters of the beta distribution used by Bernoulli TS) used during the data collection process.

[5] The dataset is licensed under CC BY 4.0.

**Open Bandit Pipeline.** To facilitate the use of OPE in practice and standardize its experimental procedures, we implement a Python software called *Open Bandit Pipeline*.[6] Our software contains the following main modules:

- The **dataset** module provides a data loader to preprocess Open Bandit Dataset and tools to generate synthetic bandit datasets. It also implements functions to handle multiclass classification datasets as bandit data, which is useful when we conduct OPE experiments in research papers.

- The **policy** module implements several online bandit algorithms and off-policy learning methods such as a neural network-based off-policy learning method. This module also implements interfaces that allow practitioners to easily evaluate their own policies in their business using OPE.

- The **ope** module implements several existing OPE estimators including DM, IPW, DR, and some advanced ones such as Switch [51], More Robust Doubly Robust (MRDR) [10], and DR with Optimistic Shrinkage (DRos) [41]. This module also provides generic abstract interfaces to support custom implementations so that researchers can evaluate their own estimators easily.

Appendix E describes how the software facilitates the evaluation of OPE and bandit algorithms. We also prepare plenty of tutorial contents at `https://github.com/st-tech/zr-obp/tree/master/examples/quickstart` to help users grasp the usage of the software easily. We also provide the thorough documentation at `https://zr-obp.readthedocs.io/en/latest/`.

Every core function of the software is tested and thus are well maintained.[7] It currently has five core contributors.[8] The active development and maintenance will continue over a long period. Users can follow our progress at the following mailing list: `https://groups.google.com/g/open-bandit-project`.

We believe that the software allows researchers to focus on building their OPE estimator and to easily compare it with other methods in a standardized manner. It will also help practitioners implement cutting-edge estimators in their applications and improve their decision making systems.

## 4 Related Work

Here, we summarize the existing related work and resources, and clarify the advantages of ours.

**Related Datasets.** Our dataset is closely related to those of [24] and [25]. Lefortier et al. [24] introduce a large-scale logged bandit data (Criteo Data[9]) from a leading company in display advertising, Criteo. The dataset contains context vectors of user impressions, advertisements (ads) as actions, and click indicators as rewards. It also provides the ex-ante probability of each ad being selected by the behavior policy. Therefore, this dataset can be used to compare different off-policy *learning* methods, which aim to learn a new policy using only logged bandit data. In contrast, Li et al. [25] introduce a dataset (Yahoo! R6A&B[10]) collected on a news recommendation interface of the Yahoo! Today Module. The dataset contains context vectors of user impressions, presented news as actions, and click indicators as rewards. The dataset was collected by running a uniform random policy on the news recommendation platform, allowing researchers to evaluate their (online or offline) bandit algorithms.

However, the existing datasets have several limitations, which we overcome as follows:

- The existing datasets include only a single logged bandit dataset collected by running only a single policy. Moreover, the previous datasets do not provide the implementation to replicate the policies used during data collection. As a result, these datasets cannot be used for

---

[6]https://github.com/st-tech/zr-obp

[7]https://github.com/st-tech/zr-obp/tree/master/tests

[8]https://github.com/st-tech/zr-obp/graphs/contributors

[9]https://www.cs.cornell.edu/ãdith/Criteo/

[10]`https://webscope.sandbox.yahoo.com/catalog.php?datatype=r`

Table 2: Comparison of currently available bandit datasets

|  | Criteo Data [24] | Yahoo! R6A&B [25] | Open Bandit Dataset (ours) |
|---|---|---|---|
| **Domain** | Display Advertising | News Recommendation | Fashion E-Commerce |
| **Dataset Size** | ≈ 103M | ≈ 40M | ≈ 26M |
| **#Data Collection Policies** | 1 | 1 | **2** |
| **Uniform Random Data** | ✗ | ✔ | ✔ |
| **Data Collection Policy Code** | ✗ | ✗ | ✔ |
| **Evaluation of Bandit Algorithms** | ✔ | ✔ | ✔ |
| **Evaluation of OPE** | ✗ | ✗ | ✔ |
| **Pipeline Implementation** | ✗ | ✗ | ✔ |

*Note*: **Dataset Size** is the total number of samples included in the whole dataset. **#Data Collection Policies** is the number of policies that were used to collect the data. **Uniform Random Data** indicates whether the dataset contains a subset of data generated by the uniform random policy. **Data Collection Policy Code** indicates whether the code to replicate data collection policies is publicized. **Evaluation of Bandit Algorithms** indicates whether it is possible to use the data to evaluate bandit algorithms. **Evaluation of OPE** indicates whether it is possible to use the dataset to evaluate OPE estimators. **Pipeline Implementation** indicates whether a pipeline tool to handle the dataset is available.

Table 3: Comparison of currently available packages of bandit algorithms and OPE

|  | Vowpal Wabbit [3] | contextualbandits [5] | RecoGym [36] | Open Bandit Pipeline (ours) |
|---|---|---|---|---|
| **Synthetic Data Generator** | ✗ | ✗ | ✔ | ✔ |
| **Classification Data Handler** | ✗ | ✗ | ✗ | ✔ |
| **Support for Real-World Data** | ✗ | ✗ | ✗ | ✔ |
| **Bandit Algorithms** | ✔ | ✔ | ✔ | ✔ |
| **Basic OPE Estimators** | ✔ | ✔ | ✗ | ✔ |
| **Advanced OPE Estimators** | ✗ | ✗ | ✗ | ✔ |
| **Evaluation of OPE** | ✗ | ✗ | ✗ | ✔ |

*Note*: **Synthetic Data Generator** indicates whether it is possible to generate synthetic bandit data with the package. **Classification Data Handler** indicates whether it is possible to transform multiclass classification data to bandit data with the package. **Support for Real-World Data** indicates whether it is possible to handle real-world bandit data with the package. **Bandit Algorithms** indicates whether the package includes implementations of online and offline bandit algorithms. **Basic OPE Estimators** indicates whether the package includes implementations of *basic* OPE estimators such as DM, IPW, and DR described in Section 2. **Advanced OPE Estimators** indicates whether the package includes implementations of *advanced* OPE estimators such as Switch and More Robust Doubly Robust described in Appendix B. **Evaluation of OPE** indicates whether it is possible to evaluate the accuracy of OPE estimators with the package.

the comparison of different OPE estimators, although they can be used to evaluate policy *learning* methods.

→ In contrast, we provide the code to replicate the data collection policies in our software, which allows researchers to rerun the same policies on the log data. Without the code of the exact algorithms, we could not implement the evaluation of OPE experiments. Therefore, the code and algorithm release is an essential component of our open-source. Moreover, our dataset consists of a set of *two* different logged bandit datasets generated by running two different policies on the same platform. It enables the comparison of different OPE estimators, as we show in Section 5.

• The existing datasets do not provide a tool to handle their data. Researchers have to reimplement the experimental environment by themselves before implementing their own OPE estimators. This can lead to inconsistent experimental conditions across different studies, potentially causing reproducibility issues.

→ We implement Open Bandit Pipeline to simplify and standardize the experimental processing of bandit algorithms and OPE. This tool thus contributes to the reproducible and transparent use of our dataset.

Table 2 summarizes the key differences between our dataset and the existing ones.

**Related Packages.**    There are several existing packages related to Open Bandit Pipeline. *Vowpal Wabbit*[11] is a library for fast machine learning, online learning, contextual bandits, and reinforcement learning [3]. It handles learning problems with any number of sparse features, achieving great scaling. The *contextualbandits* package[12] contains implementations of several contextual bandit algorithms [5]. It aims to provide an easy procedure to compare bandit algorithms to reproduce research papers that do not provide easily available implementations. There is also *RecoGym*[13], which focuses on providing simulation bandit environments imitating the e-commerce recommendation setting [36].

However, the following features differentiate our software from the previous ones:

- The previous packages focus on implementing and comparing online bandit algorithms or off-policy learning methods. However, they ***cannot*** be used to implement several advanced OPE estimators. They also do not help conduct benchmark experiments of OPE estimators.
  → Our software implements a wide variety of OPE estimators, including advanced ones such as Switch, MRDR, and DRos. In addition, Open Bandit Pipeline provides estimators that can address continuous actions or combinatorial actions, which cannot be handled by other packages. It also provides flexible interfaces for implementing new OPE estimators, allowing researchers to plug in their own estimators and compare them with existing estimators easily.

- The previous packages accept their own interface and data formats. Thus, they are not user-friendly.
  → Our software follows the prevalent *scikit-learn* style interface and provides sufficient example codes at https://github.com/st-tech/zr-obp/tree/master/examples so that anyone, including practitioners and students, can follow the usage.

- The previous packages cannot handle real-world bandit datasets.
  → Our software comes with Open Bandit Dataset and includes the **dataset module**. This facilitates the evaluation of bandit algorithms and OPE estimators using real-world data.

Table 3 summarizes the key differences between our software and the existing ones.

**Related Benchmarks.**    There are several studies conducting benchmark experiments on OPE estimators. Fu et al. [11] provide the Deep Off-Policy Evaluation (DOPE) benchmark, which is designed to evaluate the performance of OPE estimators on several control tasks. The notable contribution of DOPE is that it evaluates the OPE performance across a range of evaluation policies with different policy values and measures performance on ranking and selection as well as policy evaluation. Evaluating the ranking and policy selection performance is challenging with real-world bandit data due to the difficulty of deploying many policies during data collection. Voloshin et al. [50] provide a benchmark study on a variety of tasks ranging from tabular problems to image-based tasks in Atari. A wide variety of OPE estimators are compared in the benchmark, including recent variants of DR such as MRDR and Self-Normalized DR (SNDR), producing a holistic summary of the challenges one should address in OPE applications.

Our work differentiates itself from these previous benchmark studies in several ways. First, our benchmark and implementation cover relevant methods that are not included in the previous benchmarks. For example, we evaluate some advanced estimators such as Switch Doubly Robust (Switch-DR), and DRos, which are not compared in DOPE. Moreover, we evaluate how a recent hyperparameter tuning method proposed in [41] works with real-world bandit data. Tuning hyperparameters of OPE estimators is a critical component in OPE application, as it can greatly affect the OPE performance. However, the previous benchmarks do not evaluate the performance of the tuning method. Moreover, we release the real-world bandit dataset that allows benchmarking of OPE estimators in a realistic scenario. Our public dataset makes it possible to identify what matters in applying OPE to real-world scenarios. This is in contrast to the previous benchmarks using only synthetic environments. Indeed, we found some critical bottlenecks, which have not yet been pointed out in the literature. Specifically, we found that it is necessary to develop a reliable method to choose and tune OPE estimators in a data-driven manner (discussed in Sections 5 and 6). We would also emphasize that the previous

---

[11]https://github.com/VowpalWabbit/vowpal_wabbit

[12]https://github.com/david-cortes/contextualbandits

[13]https://github.com/criteo-research/reco-gym

benchmarks focus on conducting comprehensive empirical studies. They do not provide implementations that allow researchers to add their own estimators to the benchmark, generate synthetic data, and handling real bandit data.

## 5 Benchmark Experiments

We perform benchmark experiments of OPE estimators using Open Bandit Dataset and Pipeline. We first describe an experimental protocol to evaluate OPE estimators and use it to compare a wide variety of existing estimators. We then discuss the initial findings from the experiments. We share the code to replicate the benchmark experiments at `https://github.com/st-tech/zr-obp/tree/master/benchmark/ope`.

### 5.1 Experimental Protocol

We can empirically evaluate OPE estimators' performance by using two sources of logged bandit data collected by running two different policies. In the protocol, we regard one policy as behavior policy $\pi_b$ and the other as evaluation policy $\pi_e$. We denote log data generated by $\pi_b$ and $\pi_e$ as $\mathcal{D}^{(b)} := \{(x_i^{(b)}, a_i^{(b)}, r_i^{(b)})\}_{i=1}^{n^{(b)}}$ and $\mathcal{D}^{(e)} := \{(x_i^{(e)}, a_i^{(e)}, r_i^{(e)})\}_{i=1}^{n^{(e)}}$. Then, by applying the following protocol to several different OPE estimators, we compare their estimation performance:

1. Estimate the policy value of $\pi_e$ using $\mathcal{D}^{(b)}$ by OPE estimator $\hat{V}$. We represent a policy value estimated by $\hat{V}$ as $\hat{V}(\pi_e; \mathcal{D}^{(b)})$.

2. Evaluate the estimation accuracy of $\hat{V}$ using the following *squared error* (SE):

$$SE(\hat{V}; \mathcal{D}^{(b)}) := \left( \hat{V}(\pi_e; \mathcal{D}^{(b)}) - V_{\mathrm{on}}(\pi_e) \right)^2,$$

where $V_{\mathrm{on}}(\pi_e) := (1/n^{(e)}) \sum_{i=1}^{n^{(e)}} r_i^{(e)}$ is the Monte-Carlo estimate (on-policy estimate) of $V(\pi_e)$ based on $\mathcal{D}^{(e)}$.

3. Repeat the above process $T$ times with different bootstrap samples and calculate the following *root mean-squared-error* (RMSE) as the estimators' performance measure.

$$RMSE(\hat{V}; \mathcal{D}^{(b)}) := \sqrt{\frac{1}{T} \sum_{t=1}^{T} SE(\hat{V}; \mathcal{D}_t^{(b,*)})},$$

where $\mathcal{D}_t^{(b,*)}$ is the $t$-th bootstrapped sample of $\mathcal{D}^{(b)}$.

Algorithm 1 in Appendix C describes the experimental protocol to evaluate OPE estimators in detail.

### 5.2 Compared Estimators

We compare the following OPE estimators: DM, IPW, Self-Normalized Inverse Probability Weighting (SNIPW), DR, SNDR, Switch-DR, and DRos. We tune the built-in hyperparameter of Switch-DR and DRos using a data-driven hyperparameter tuning method described in Su et al. [41]. The details of the above estimators and data-driven hyperparameter tuning method are given in Appendix B.

For estimators except for DM, we use the true action choice probability $\pi_b(a|x)$ contained in Open Bandit Dataset. For estimators except for IPW and SNIPW, we need to obtain a reward estimator $\hat{q}$. We do this by using gradient boosting[14] (implemented in *scikit-learn* [34]) and training it on $\mathcal{D}^{(b)}$. We also use cross-fitting [19, 33] to avoid substantial bias from overfitting when obtaining $\hat{q}$.

### 5.3 Results

The results of the benchmark experiments with $n = 300,000$ are given in Table 4. We describe **Bernoulli TS $\rightarrow$ Random** to represent the OPE situation where we use Bernoulli TS as $\pi_b$ and Random as $\pi_e$. Please see Appendix C for additional results.

---

[14]Specifically, we use 'sklearn.ensemble.HistGradientBoostingClassifier(learning_rate=0.01, max_iter=100, max_depth=5, min_samples_leaf=10, random_state=12345)' to obtain $\hat{q}$.

Table 4: RMSE ($\times 10^3$) of OPE estimators (**Bernoulli TS $\rightarrow$ Random**)

| OPE Estimators ($\hat{V}$) | ALL | Men's | Women's |
|---|---|---|---|
| **IPW**[1] | 0.493 (1.560)[3] | 0.789 (1.719) | 0.776 (1.382)[3] |
| **SNIPW**[2] | 0.507 (1.602)[3] | 0.644 (1.403)[1/3] | 0.804 (1.433)[3] |
| **DM**[3] | **1.026 (3.244)** | 0.773 (1.685) | **0.816 (1.455)** |
| **DR**[4] | 0.482 (1.526)[3] | 0.613 (1.336)[1/3] | 0.803 (1.430)[3] |
| **SNDR**[5] | 0.482 (1.526)[3] | 0.659 (1.436)[1/3] | 0.791 (1.408)[3] |
| **Switch-DR**[6] | 0.482 (1.526)[3] | 0.613 (1.336)[1/3] | 0.803 (1.430)[3] |
| **DRos**[7] | **0.316 (1.000)**[1/2/3/4/5/6] | **0.459 (1.000)**[1/2/3/4/5/6] | **0.561 (1.000)**[1/2/3/4/5/6] |

Table 5: RMSE ($\times 10^3$) of DRos with different hyperparameter values (**ALL Campaign**)

| $\lambda$ of DRos | Random $\rightarrow$ Bernoulli TS | Bernoulli TS $\rightarrow$ Random |
|---|---|---|
| **1** | 1.384 (2.906) | 0.963 (3.920) |
| **5** | 1.247 (2.619) | 0.837 (3.407) |
| **10** | 1.162 (2.439) | 0.770 (3.135) |
| **50** | 0.896 (1.881) | 0.589 (2.398) |
| **100** | 0.778 (1.634) | 0.498 (2.029) |
| **500** | 0.574 (1.206) | 0.294 (1.199) |
| **1,000** | 0.482 (1.106) | **0.245 (1.000)** |
| **5,000** | **0.476 (1.000)** | 0.270 (1.101) |
| **10,000** | **0.476 (1.000)** | 0.323 (1.315) |
| **tuning** | **0.476 (1.000)** | 0.323 (1.315) |

*Note*: Root mean squared errors (RMSEs) estimated with 200 different bootstrapped iterations are reported ($n = 300,000$). RMSEs normalized by the best (lowest) RMSE are reported in parentheses. [1/2/3/4/5/6/7] denote a significant difference compared to the indicated estimator (Wilcoxon rank-sum test, $p < 0.05$). The **red bold** is used when the best results outperform the second bests in a statistically significant level. The **blue bold** is used when the worst results underperform the second worsts in a statistically significant level. $\pi_b \rightarrow \pi_e$ represents the OPE situation where the estimators aim to estimate the policy value of $\pi_e$ using logged bandit data collected by $\pi_b$.

**Performance comparisons.** Table 4 shows that DRos (with automatic hyperparameter tuning) performs best for the three campaigns, achieving about 30-60% more accurate OPE than the second-best estimators. We then evaluate several values for the hyperparameter $\lambda$ of DRos. Table 5 shows the OPE performance (RMSE) of DRos with different values of $\lambda$ ($\in \{1, 5, 10, 50, \ldots, 10000\}$). This table also includes the OPE performance of DRos with automatic hyperparameter tuning of Su et al. [41]. First, we observe that the choice of $\lambda$ greatly affects the performance of DRos. Specifically, for **Random $\rightarrow$ Bernoulli TS**, a larger value of $\lambda$ leads to a better OPE performance. In contrast, for **Bernoulli TS $\rightarrow$ Random**, $\lambda = 1,000$ is the best setting. Second, we observe that the automatic hyperparameter tuning procedure prefers a large value of $\lambda$. This means that the tuning procedure puts emphasis on the bias of the estimator, as a large value of $\lambda$ leads to a low bias, but a high variance estimator. This strategy succeeds for **Random $\rightarrow$ Bernoulli TS**. However, we observe that there is room for improvement for **Bernoulli TS $\rightarrow$ Random** in terms of automatic hyperparameter tuning. This suggests opportunities for further investigations on the quality of automatic hyperparameter tuning for achieving more accurate OPE in practice.

**OPE performance with different sample sizes.** Next, we compare the *small*-sample setting ($n = 10,000$) and *large*-sample setting ($n = 300,000$) to evaluate how the OPE performance changes with the sample size. We observe in Table 6 that the estimators' performance can change significantly depending on the size of the logged bandit data. In particular, for the Men's and Women's campaigns, the most accurate estimator changes with the sample size. The table shows that DM outperforms the other estimators in the *small*-sample setting, while DRos is the best for the *large*-sample setting. These observations suggest that practitioners have to choose an appropriate OPE estimator carefully for their specific application. It is thus necessary to develop a reliable method to choose and tune OPE estimators in a data-driven manner. Specifically, in real applications, we have to tune the estimators'

Table 6: Comparison of *small*-sample and *large*-sample OPE performance (RMSE $\times 10^3$)

| OPE Estimators | ALL | | Men's | | Women's | |
|---|---|---|---|---|---|---|
| | *small*-sample | *large*-sample | *small*-sample | *large*-sample | *small*-sample | *large*-sample |
| **IPW**[1] | 1.899 | 0.493[3] | 3.683[5] | 0.789 | **3.156** | 0.776[3] |
| **SNIPW**[2] | 1.641 | 0.507[3] | 3.661[4/5/6] | 0.644[1/3] | 3.038[1/5] | 0.804[3] |
| **DM**[3] | 0.797[1/2/4/5/6] | **1.026** | **3.041**[1/2/4/5/6/7] | 0.773 | **2.665**[1/2/4/5/6/7] | **0.816**[3] |
| **DR**[4] | 1.203[1/2] | 0.482[3] | 3.747 | 0.613[1/3] | 3.055[1/2] | 0.803[3] |
| **SNDR**[5] | 1.159[1/2] | 0.482[3] | 3.757 | 0.659[1/3] | 3.069[1] | 0.791[3] |
| **Switch-DR**[6] | 1.203[1/2] | 0.482[3] | 3.747 | 0.613[1/3] | 3.055[5] | 0.803[3] |
| **DRos**[7] | 0.765[1/2/4/5/6] | **0.316**[1/2/3/4/5/6] | 3.727 | **0.459**[1/2/3/4/5/6] | 3.051[1/5] | **0.561**[1/2/3/4/5/6] |

*Note*: Root mean squared errors (RMSEs) estimated with 200 different bootstrapped iterations are reported (**Bernoulli TS** $\to$ **Random**). $n = 10,000$ for the *small*-sample setting, while $n = 300,000$ for the *large*-sample setting. [1/2/3/4/5/6/7] denote a significant difference compared to the indicated estimator (Wilcoxon rank-sum test, $p < 0.05$). The **red bold** is used when the best results outperform the second bests in a statistically significant level. The **blue bold** is used when the worst results underperform the second worsts in a statistically significant level.

hyperparameters or identify an accurate estimator without the ground-truth or on-policy policy value of the evaluation policy.

# 6 Conclusion, Limitations, and Future Work

To enable a realistic and reproducible evaluation of OPE, we presented Open Bandit Dataset, a set of logged bandit datasets collected on a fashion e-commerce platform. The dataset comes with Open Bandit Pipeline, Python software that makes it easy to evaluate and compare different OPE estimators. We aim to facilitate understanding of the empirical properties of OPE estimators and address experimental inconsistencies in the literature. We also perform extensive experiments on a variety of OPE estimators and analyze the effects of hyperparameter choice and sample size in a real-world setting. Our experiments highlight that an appropriate estimator can change depending on a problem setting such as the sample size. The results also suggest that there is room to improve the OPE performance in terms of automatic hyperparameter tuning and estimator selection. These observations call for a new estimator selection method and a hyperparameter tuning procedure to be developed.

A limitation is that we assume that the reward of an item at a position does not depend on other simultaneously presented items. This assumption might not hold, as an item's attractiveness can have a significant effect on the expected reward of another item in the same recommendation list [27]. To address more realistic situations, we have implemented some OPE estimators for the slate action setting [30, 45] in Open Bandit Pipeline. Comparing the standard OPE estimators and those for the slate action setting on our dataset is a valuable and interesting research direction.

Open Bandit Dataset is currently the only public dataset allowing OPE experiments. Therefore, it might lead to an overfitting issue. Moreover, Open Bandit Dataset includes only two policies. Here, we emphasize that there has never been any public real-world data that allow realistic and reproducible OPE research before. Our open-source project is an initial step towards this goal. Having many policies would be even more valuable, but releasing data with two different data collection policies is distinguishable enough from the prior work. We believe that our work will inspire other researchers and companies to create follow-up benchmark datasets to advance OPE research further.

## Acknowledgments and Disclosure of Funding

We thank Haruka Kiyohara, Ryo Kuroiwa, Richard Liu, Kazuki Mogi, Masahiro Nomura, Kyohei Okumura, Ayumi Sudo, and Koichi Takayama for their thoughtful comments on the manuscript and help in developing the software. We would also like to thank anonymous reviewers for their helpful feedback.

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
