# OpenReview forum: "Open Bandit Dataset and Pipeline: Towards Realistic and Reproducible Off-Policy Evaluation"
_NeurIPS.cc/2021/Track/Datasets_and_Benchmarks/Round2 — NeurIPS 2021 Datasets and Benchmarks Track (Round 2)_

### Official Review · Reviewer_UTTX · 2021-09-19
**Review of "open bandit dataset and pipeline: towards realistic and reproducible off-policy evaluation"**

**Rating:** 6
**Confidence:** 4
**Correctness:** The claims made in the submission see…

**Strengths:**

* Beyond its utility as a publicly-available logged-bandit dataset, *Open Bandit Dataset* enables researches to evaluate OPE estimators in a realistic environment without relying on simulators. According to the authors, OBD is first publicly-available dataset to enable such evaluation. Using this new evaluation procedure, the authors were able to identify a new potential research direction, namely the need for better automatic hyper-parameter tuning methods in certain instances.

* The fact that OBD has already seen use in several publications is noteworthy.

* The dataset is well documented and its use is well supported by the accompanying software package *Open Bandit Pipeline*.

**Weaknesses:**

Although how OBD enables realistic evaluation of OPE estimators is well explained and a detailed evaluation protocol is provided, I found some other features of OBD and in particular OBP to be lacking more detailed explanations:

* It is mentioned that OBP includes tools to generate synthetic datasets as well but how OBP does so is not discussed anywhere on the paper. I believe this is achieved by "modifying classification datasets", which is mentioned in the introduction as an existing synthetic approach to evaluating OPE estimators, but I do not understand what is meant by that.

* Table 2 shows that OBD can be used to evaluate bandit algorithms (not only OPE estimators). However, what is meant by "evaluation of bandit algorithms" and how OBD can facilitate that is not discussed anywhere else.

**Additional Feedback:**

* At line 188, it is mentioned that alternative software packages cannot be used to implement some advanced OPE estimators. Does it mean that (unlike OBP) they do not provide interfaces flexible enough for implementing such advanced OPE estimators?

* At the end of Section 6, an assumption on the reward structure is mentioned as a limitation. As far as I understood, this is a limitation of the experimental results presented in the paper and not an inherit limitation of OBD. Is that correct?

**Clarity:**

The paper is generally well-written with the exception of points that I mentioned in weaknesses.

**Documentation:**

The dataset is easily accessible and well documented. The authors mention that they aim to keep mantaining the dataset but I have not seen a detailed maintenance plan.

**Ethics:**

There are no ethical concerns.

**Relation To Prior Work:**

Both related datasets as well as related software packages are summarized in Tables 2 and 3. These tables makes it clear how OBD and OBP differ from prior work.

**Summary And Contributions:**

The authors introduce a new logged-bandit dataset called *Open Bandit Dataset*. This new dataset consists of data collected in the same environment by two different policies hence it enables the realistic evaluation of OPE estimators (by comparing the on-policy performance of one policy with the off-policy estimation of its performance obtained using the data collected by the other policy). The authors also introduce *Open Bandit Pipeline*, which is a software package that consists of modules for dataset preprocessing, policy learning, and OPE estimation.

---

> ### Author Response · Authors · 2021-09-24
> **Author response**
>
> Thank you for your useful, detailed feedback. We will update the paper with the suggested minor revisions and respond below to some concrete questions and comments.
>
> > It is mentioned that OBP includes tools to generate synthetic datasets as well but how OBP does so is not discussed anywhere on the paper. I believe this is achieved by "modifying classification , datasets", which is mentioned in the introduction as an existing synthetic approach to evaluating OPE estimators, but I do not understand what is meant by that.
>
> Thank you for raising this point.  OPB supports OPE experiments with “fully synthetic datasets” and “modified classification datasets”. “fully synthetic datasets” do not use classification data and (context, action, reward) are all sampled from synthesized functions and distributions. “modified classification datasets” are created from multi-class classification datasets. In OPE, it is common to transform classification datasets to logged bandit data to compare the performance of OPE estimators in a synthetic manner. [6, 8, 16, 34, 41, 44] rely on “modified classification datasets” to conduct OPE experiments. Appendix G of [8, https://arxiv.org/pdf/1802.03493.pdf ] describes how to modify classification datasets into bandit data in detail.
>
> These procedures  are not described in the paper in detail, but we prepare a lot of tutorials as to how to implement OPE experiments in a synthetic manner in the package repository. For example,
>
> https://github.com/st-tech/zr-obp/blob/master/examples/quickstart/synthetic.ipynb
> explains how to implement OPE experiments with fully synthetic data and
>
> https://github.com/st-tech/zr-obp/blob/master/examples/quickstart/multiclass.ipynb
> explains how to implement OPE experiments by modifying classification data to bandit data.
>
> In the paper, we focused on OPE experiments with real-world data, because we think this point differentiates our resources from others the most. However, as we described here, OBP is not limited to experiments with real-world data. Researchers can also accelerate their research using synthetic data or classification data with OBP.
>
> > Table 2 shows that OBD can be used to evaluate bandit algorithms (not only OPE estimators). However, what is meant by "evaluation of bandit algorithms" and how OBD can facilitate that is not discussed anywhere else.
>
> Thank you for the important point. Researchers working on bandit algorithms want to evaluate the performance of their proposed bandit algorithms with real-world data. “evaluation of bandit algorithms” is to evaluate or compare the performance of different bandit algorithms (such as UCB or Thompson Sampling). For example, one recent paper (https://arxiv.org/abs/2107.11419 , [18]) uses OBD for the “evaluation of bandit algorithms”.
>
> In our paper, we focused on the OPE side, as there are some other resources that enable “evaluation of bandit algorithms” as we summarized in Tables 2 and 3. We think that benchmarking OPE estimators in a real-world scenarios is the most unexplored direction.
>
> > At line 188, it is mentioned that alternative software packages cannot be used to implement some advanced OPE estimators. Does it mean that (unlike OBP) they do not provide interfaces flexible enough for implementing such advanced OPE estimators?
>
> Exactly. For example, when we propose a new OPE estimator, we first implement it and then compare it with other existing estimators. We can do this process easily with OBP, because it allows us to implement a new estimator using our flexible interface and immediately compare it with a variety of existing estimators that are already implemented in the package. Other related packages do not implement many of the existing OPE estimators nor provide us an interface to add a new estimator.
>
> > At the end of Section 6, an assumption on the reward structure is mentioned as a limitation. As far as I understood, this is a limitation of the experimental results presented in the paper and not an inherit limitation of OBD. Is that correct?
>
> Yes, you are right. The assumption on the reward structure we used in the experiments is one of the popular assumptions in the OPE literature [22]. However, we think that we can relax the assumptions. We are now implementing some new estimators that are based on more relaxed assumptions. Then, we will be able to compare a wider variety of estimators. The most recent pull request in this direction is below (showing our effort towards this point):
>
> https://github.com/st-tech/zr-obp/pull/133

---

> > ### Author Response · Authors · 2021-09-26
> > **The manuscript is updated to reflect the review**
> >
> > Thank you again for your thoughtful and in-depth review. As we are allowed to upload an updated version of the manuscript and add an additional (10th) page during the discussion period, we updated the package usage section based on your comments.
> >
> > Specifically, **we have added some additional example codes to Appendix E**. The additional examples include **“OPE experiment using fully synthetic data”**, **“OPE experiment using multiclass classification data”**, and **“Bandit experiment using Open Bandit Data”**. We also mention, in the main text, that Appendix E now contains those example codes (p4, l138). We believe that the additional contents will address your concerns (weaknesses) and clarify what "modifying classification datasets" and "evaluation of bandit algorithms" mean.
> >
> > Moreover, as we described in the previous response, many tutorial implementations related to those topics are already available on the GitHub repository. For example, we prepare the following example scripts for “evaluation of bandit algorithms” as follows.
> >
> > - **evaluation of online bandit algorithms**
> >     - (quickstart) https://github.com/st-tech/zr-obp/blob/master/examples/quickstart/online.ipynb
> >     - ​(experimental workflow) https://github.com/st-tech/zr-obp/tree/master/examples/online
> >
> > - **evaluation of offline bandit algorithms**
> >     - (quickstart) https://github.com/st-tech/zr-obp/blob/master/examples/quickstart/opl.ipynb
> >     - (experimental workflow) https://github.com/st-tech/zr-obp/tree/master/examples/opl

---

### Official Review · Reviewer_CpZg · 2021-09-19
**Review of the Off Policy Evaluation Benchmark**

**Rating:** 6
**Confidence:** 2

**Strengths:**

The paper claims to be the first to propose a large realistic dataset for off policy evaluation in the bandit setting. They provide an easy to use codebase and have implemented a number of baselines for their dataset which would lead to ease of future research on this benchmark.

**Weaknesses:**

The motivation for a large-scale bandit based dataset is not clear. There has been a lot of work done on bandits on simpler settings and its not clear what this new dataset brings to the table. On the other end of the spectrum, the authors do not provide a discussion with other related work like Fu et. al (2021): Benchmarks for Deep Off-Policy Evaluation which proposes a benchmark for off-policy evaluation. Given that there exist benchmarks for complex off-policy evaluation in a non-bandit setting as well as simpler benchmarks in the bandit setting where one could run as many policies as possible, its not immediately clear if the proposed benchmark provides a significantly different contribution.

**Additional Feedback:**

The authors provide a dataset for off policy evaluation in bandit models. The motivation and writing could use a bit of work since its not immediately clear how the benchmark provides something significantly new and needed in the field.

**Update after Rebuttal**: I have increased the score to 6 based on the points raised by the authors during the rebuttal.

**Clarity:**

The paper is a bit difficult to follow and restructuring the sections would allow for an easier read. In particular, details regarding sklearn and the codebase can be provided in the Appendix and using the main paper to provide more information on the baselines would serve better.

**Correctness:**

The benchmark is constructed in a sound way and the evaluation methods and experiment design makes sense.

**Documentation:**

The authors provide thorough documentation of their dataset as well as the different algorithms evaluated in the benchmarking. The details on how to use the pipeline are clearly provided and they do a pretty good job at making the whole benchmark easy to set up and experiment with.

**Ethics:**

I do not see any ethical concerns regarding the work.

**Relation To Prior Work:**

The paper discusses how its contributions are different when compared with some of the other benchmarks on recommender systems and bandit problems. There is little discussion on previous works on off policy evaluation benchmarks from the different sectors (eg. healthcare or deep off policy evaluation).

**Summary And Contributions:**

The paper proposes a new dataset called ZOZOTOWN with the aim of benchmarking off policy evaluation methods for the bandit setting. They collect a massive amount of data through two different policies and then evaluate different algorithms on their dataset. They also provide their codebase pipeline for basic usage of their dataset.

---

> ### Author Response · Authors · 2021-09-24
> **Author response**
>
> Thank you for your useful, detailed feedback. We will update the paper with the suggested minor revisions and respond below to some concrete questions and comments.
>
> > The motivation for a large-scale bandit based dataset is not clear. There has been a lot of work done on bandits on simpler settings and its not clear what this new dataset brings to the table. On the other end of the spectrum, the authors do not provide a discussion with other related work like Fu et. al (2021): Benchmarks for Deep Off-Policy Evaluation which proposes a benchmark for off-policy evaluation.
>
> Thank you for pointing this out. We agree that we should cite the paper you mentioned (Fu et. al (2021)) as related work. At the same time, we would emphasize that their and other previous studies all rely on fully synthetic environments. We cannot evaluate and compare how OPE estimators work in realistic scenarios. This drawback of the literature makes it difficult to identify bottlenecks of the existing estimators, what aspect matters in applying OPE to the real-world, and which estimators to use in practice.
>
> The other two reviewers acknowledge the value of our datasets and packages that allow researchers to evaluate the OPE performance in a real-world environment (not synthetic environment as the previous works). Indeed, we found some critical bottlenecks that are important in applying OPE to the real-world, which have not yet been pointed out in the literature. Specifically, we found that it is necessary  to develop a reliable method to choose and tune OPE estimators in a data-driven manner (i.e., data driven hyperparameter tuning and estimator selection), as discussed in Sections 5 and 6. We can specify these bottlenecks that matter in the real-world, as we rely on the real-world resource in our benchmark. We believe that this point differentiates our resource from the previous synthetic benchmarks.

---

> > ### Author Response · Authors · 2021-09-26
> > **The manuscript is updated to reflect the review**
> >
> > Thank you again for your thoughtful and in-depth review. As we are allowed to upload an updated version of the manuscript and add an additional (10th) page during the discussion period, we updated the related work section based on your comments. Specifically, we have added a paragraph titled “Related Benchmarks” in Section 4 and included Fu et al. [10] and Voloshin et al. [46] as related benchmarks.
> >
> > In summary, our benchmark study offers some advantages over those previous benchmarks (please see the updated part in Section 4 for a more detailed discussion).
> >
> > First, our benchmark and implementation cover relevant methods that are not included in the previous benchmarks. For example, we evaluate some advanced estimators such as Switch Doubly Robust (Switch-DR), and DRos, which are not compared in DOPE. Moreover, we evaluate how a recent hyperparameter tuning  method proposed in [37] works with real-world bandit data. Both previous benchmarks do not evaluate the performance of the tuning method.
> >
> > In addition, our real-world dataset makes it possible to identify what matters in applying OPE to real-world scenarios. This is in contrast to the previous benchmarks using only synthetic environments.
> >
> > Let us also empathize that our contribution is not limited to the benchmark results. The main contribution of our work is the dataset and package release. We have released Open Bandit Dataset, the large-scale real-world bandit dataset enabling the evaluation of OPE for the first time with **real-world** data. Moreover, we implemented Open Bandit Pipeline that allows researchers to add their own estimators to the benchmark easily while ensuring the reproducibility of their results. It also implements a wide variety of estimators in a user-friendly manner making it easier to apply OPE to real-world problems. We believe that the dataset and package are our unique contributions, differentiating our work from the previous benchmark-focus studies.

---

> > > ### Comment · Reviewer_CpZg · 2021-09-26
> > > **Reviewer Response**
> > >
> > > Thank you for addressing some of the concerns raised. In light of this, I have decided to increase the score to 6.

---

### Official Review · Reviewer_EaSH · 2021-09-20

**Rating:** 6
**Confidence:** 4
**Correctness:** The paper is correct for the most part.
**Clarity:** The paper is clear and well written.

**Strengths:**

In the beginning, I really enjoyed the paper and completely resonated with it -- specifically, the lack of open-source benchmarks for contextual bandits and off-policy estimation and learning. I think this is the biggest strength of this work. While the dataset will be useful, my only reservation is in the experiments (see more below). Having worked on contextual bandits in the past, I understand the importance of this work. However, there were a few aspects of the evaluation that I was disappointed with (again, see below).

**Weaknesses:**

As mentioned above, I was very excited about this paper in the beginning and completely resonated with the motivation. However, there are several limitations which I noticed towards the end, leaving me a little disappointed:

- The authors have no mention of vowpal wabbit for off-policy estimation and learning. This is very related work
- The authors only focus on off-policy estimation but not learning/optimization. It is not clear why. I believe the authors should also look into optimizing the contextual bandit policies. Vowpal wabbit, for example, provides a number of algorithms for off-policy learning. Given the logged data, why not use it to learn policies?
- I will also encourage the authors to take a look at the policy optimizer for exponential models (POEM) approach which can do a soft-max relaxation to the policy optimization problem.
- Again, to summarize, I think a key limitation of this work (in my opinion -- again, it is totally possible I could have misunderstood aspects of this work) is that it does not do the policy optimization which I think is critical in this benchmarking. Which kind of policies works best? How much does exploration hurt the performance?

**Additional Feedback:**

See issues raised above

**Documentation:**

It seems to be well documented

**Ethics:**

No ethics issues

**Relation To Prior Work:**

The paper misses some related work (mainly, for example, VW and the POEM approach for policy optimization)

**Summary And Contributions:**

This paper presents a open bandit dataset for evaluating contextual bandit algorithms. Overall, I think this paper is addressing a missing need since one of the key limitations of academic research in contextual bandits is the lack of an open-source dataset to evaluate algorithms that is not synthetically designed or obtained from classification datasets.

---

> ### Author Response · Authors · 2021-09-24
> **Author response**
>
> Thank you for your useful, detailed feedback. We will update the paper with the suggested minor revisions and respond below to some concrete questions and comments.
>
> > The authors have no mention of vowpal wabbit for off-policy estimation and learning. This is very related work
>
> Thank you for pointing this out. We agree that we should add Vowpal Wabbit (VW) to the list of related work (Table 3). An important distinction we want to emphasize here is that VW does not implement advanced OPE estimators, Evaluation of OPE protocol, and functions to handle real-world bandit data. Thus, we would argue that VW is similar to other related packages we have already mentioned (i.e., contextualbandits [3] and RecoGym [31]). Moreover, one major issue with VW is that it is based on a specific data format and not easy to adapt to its usage. In contrast, our OBP package is user-friendly and it works almost identically for the widely used scikit-learn package. (You can see how OBP works in our tutorials such as https://github.com/st-tech/zr-obp/blob/master/examples/quickstart/synthetic.ipynb )
>
> > The authors only focus on off-policy estimation but not learning/optimization. It is not clear why. I believe the authors should also look into optimizing the contextual bandit policies. Vowpal wabbit, for example, provides a number of algorithms for off-policy learning. Given the logged data, why not use it to learn policies?
>
> > I will also encourage the authors to take a look at the policy optimizer for exponential models (POEM) approach which can do a soft-max relaxation to the policy optimization problem.
>
> Thank you for raising this important point. We focused on the OPE side because OPE experiments and implementations most clearly differentiate our resource from the other existing resources. There is no real-world data and package that allow realistic and reproducible OPE experiments, making our OPE benchmark a valuable first step towards understanding the behavior of OPE estimators in real-world environments and to specify the bottleneck of the current state-of-the-arts. Indeed, we found some critical bottlenecks (i.e., data driven hyperparameter tuning and estimator selection) that are important in applying OPE to the real-world, which have not yet been pointed out in the literature.
>
> As you mentioned, it is also possible to benchmark bandit algorithms and off-policy learning methods (such as POEM) using our dataset and package. However, this has already been done with the Criteo data [19]. Therefore, we thought that conducting comprehensive experiments on the OPE estimators is the most unexplored direction. While we believe that the current benchmark experiments provide valuable contributions to the broad OPE literature, we will also work on the policy learning side. More specifically, we can compare several OPE estimators as an objective function in off-policy learning. We can also compare online contextual bandit algorithms such as Linear UCB and Thompson sampling. What kind of regularization (e.g., variance regularization in [35], self-normalization in [36], and policy imitation regularization in [https://arxiv.org/abs/1901.04723 ]) is effective for off-policy learning is also an interesting and unexplored direction.

---

> > ### Author Response · Authors · 2021-09-26
> > **The manuscript is updated to reflect the review**
> >
> > Thank you again for your thoughtful and in-depth review. As we are allowed to upload an updated version of your manuscript and add an additional (10th) page during the discussion period, we updated the related work section based on your comments. Specifically, we have added “Vowpal Wabbit” (VW) to Table 3 and “Related Packages” in Section 4.
> >
> > We would empathize here that VW focuses more on online learning, contextual bandits, and large-scale machine learning, while our focus is OPE. VW does support some basic OPE estimators (https://vowpalwabbit.org/tutorials/off_policy_evaluation.html ), however, Open Bandit Pipeline implements much wider variety of OPE estimators including Switch-DR, SNDR, MRDR, DRos, and hyperparameter tuning method for those estimators. Moreover, our package supports OPE estimators to handle continuous actions (https://github.com/st-tech/zr-obp/blob/master/obp/ope/estimators_continuous.py ) and combinatorial actions (https://github.com/st-tech/zr-obp/blob/master/obp/ope/estimators_slate.py ), the advanced settings that often occur in real-world applications of OPE. Finally, VW cannot produce synthetic data, transform classification data to bandit data (for research purposes), and handle real-world bandit data. In contrast, our package provides a variety of data generating and handling functions as well as standardized experimental procedure for OPE research, which helps researchers accelerate their research while ensuring the reproducibility of their experimental results.

---

### Decision · Program_Chairs · 2021-10-11

**Decision:**

Accept

**Comment:**

This scores for this paper are still marginal. However, the authors have addressed and responded to many of the issues raised by reviewers including expanding a related work section to include relevant prior literature in this area. After considering the author rebuttal and discussing with another AC, and given this important problem space for which quality evaluation datasets would be valuable, I recommend acceptance.